# CoP/EEBP/N-FLGS Nanocomposite as an Efficient Electrocatalyst of Hydrogen Evolution Reaction in Alkaline Media

Valerii K. Kochergin [1,*], Alexander S. Kotkin [1], Roman A. Manzhos [1], Alexander G. Krivenko [1], Igor I. Khodos [2] and Eugene N. Kabachkov [1,3,*]

1    Federal Research Center of Problem of Chemical Physics and Medicinal Chemistry, Russian Academy of Sciences, Acad. Semenov Ave. 1, 142432 Chernogolovka, Russia; krivenko@icp.ac.ru (A.G.K.)
2    Institute of Microelectronics Technology and High Purity Materials, Russian Academy of Sciences, 142432 Chernogolovka, Russia
3    Institute of Solid State Physics, Russian Academy of Sciences, 2 Academician Osipyan Str., 142432 Chernogolovka, Russia
*    Correspondence: kochergin@icp.ac.ru (V.K.K.); en.kabachkov@gmail.com (E.N.K.)

**Abstract:** The search for new hydrogen evolution reaction (HER) electrocatalysts with lower cost and higher activity and stability than noble metal catalysts is essential. In this regard cobalt phosphide is considered one of the most promising nanomaterials. The present work proposes a simple and efficient method for the synthesis of a nanocomposite of graphene–phosphorene structures decorated with CoP nanoparticles 2–5 nm in size via the electrochemical exfoliation of black phosphorus carried out in the presence of nitrogen-doped few-layer graphene structures and followed by solvothermal synthesis in a $Co^{2+}$-containing solution. The obtained CoP/EEBP/N-FLGS nanocomposite demonstrates high electrocatalytic activity and stability towards HER in an alkaline medium. The nanocomposite is characterized by an overpotential of 190 mV at a current density of 10 mA cm$^{-2}$ as well as a small Tafel slope (78 mV dec$^{-1}$). These characteristics make the CoP/EEBP/N-FLGS nanocomposite superior to most electrocatalysts based on cobalt phosphides. The results of this study could be in demand for the future design and improvement of HER electrocatalysts.

**Keywords:** phosphorene; few-layer graphene structures; cobalt phosphides; nanocomposites; electrochemical exfoliation of graphite; hydrogen evolution reaction; electrocatalysis



## 1. Introduction

To date, traditional fossil fuels, such as coal, oil and natural gas, are mainly used for energy production. However, the excessive use of such fuels leads to environmental pollution and the depletion of their resources. Growing energy demands and environmental concerns highlight the importance of research into sustainable and environmentally friendly energy conversion and storage technologies [1–3]. A transition to the hydrogen power industry is believed to be the most prominent direction in the energy sector [4]. Hydrogen is treated as a promising "green" fuel, since there is no harmful emission during its combustion and the amount of resources for its production is practically unlimited. At the same time, the calorific value of hydrogen (ca. 140 kJ/g) is almost three times higher than that of gasoline. In addition, $H_2$ is also used for the production of ammonia, oil refining and other important modern industries [5]. To date, there are three methods of hydrogen production, namely the steam reforming of methane or natural gas, coal gasification and water electrolysis, the most attractive method [6]. In the first two cases, fossil fuels are used, and during hydrogen production, a large amount of environmentally harmful by-products is released. In the third case, the source of hydrogen is water, and the products of electrolysis are only $H_2$ and $O_2$ formed as a result of the hydrogen evolution reaction (HER) and oxygen evolution reaction (OER), respectively. However, the practical application water

electrolysis for hydrogen production is largely hampered due to the limitations associated with HER electrocatalysts [7,8]. Today, materials based on Pt nanoparticles and Pt group metals with high catalytic activity [9,10] are widely used as HER catalysts [11]. However, limited resources, high cost, and lack of stability hinder the large-scale use of such catalysts and, as a result, limit the hydrogen production by the electrolysis of water [12,13]. These circumstances stimulate the development of a new generation of HER catalysts with a lower cost and higher activity and stability than those of Pt-based catalysts [14,15].

Among the wide variety of electrocatalysts being developed, transition metal phosphides are the most promising materials in terms of activity, stability and cost, wherein the greatest attention is paid to nanomaterials based on cobalt phosphides [16–18]. In recent years, great efforts have been made to improve their electrocatalytic characteristics, i.e., to enhance the conductivity, to optimize the electronic structure, to increase the electrochemically active surface area (ECSA) as well as to find the additions preventing phosphide agglomeration in the composites [19,20]. Moreover, the cobalt phosphide synthesis often utilizes hypophosphite ions, which lead to the formation of highly toxic phosphine in some way [21]. To solve this problem, black phosphorus (BP), a material that became popular in the last decade, can be considered as a source of phosphorus [22]. BP is nontoxic and the most thermodynamically stable allotropic modification of phosphorus. One of the typical features of black phosphorus is a flexible variability of the band gap (0.3–2.0 eV), which depends on the number of layers [23,24]. Additionally, BP is characterized by its high carrier mobility (1000 cm$^2$ V$^{-1}$ s$^{-1}$), moderate current on/off ratio ($10^4$–$10^5$), mechanical flexibility, etc. [25]. From a structural point of view, black phosphorus is a layered substance, in which every layer is a "corrugated" sheet consisting of two-layered six-membered rings. Due to the sp$^3$ hybridization, in a vertical projection, these sheets look like honeycomb, graphene-like structures which are held in the BP crystal by weak Van der Waals forces. Black phosphorus can be easily exfoliated into several layers or a phosphorene monolayer. Similar to the production of other 2D nanomaterials, phosphorene structures can be obtained by two approaches, namely a top-down (e.g., mechanical exfoliation and liquid exfoliation) and a bottom-up approach (e.g., chemical vapor deposition and wet-chemistry synthesis) [26]. It should be emphasized that top-down methods are currently more commonly used in the production of phosphorene structures, since they have many advantages, such as low cost, simplicity, scalability, etc. Various methods of BP exfoliation into phosphorene are described in detail in [27], among which a great deal of attention is paid to sonication, anodic and cathodic electrochemical exfoliation, and bipolar electrochemical exfoliation. Electrochemically exfoliated black phosphorus (EEBP) is being studied as a potential electrocatalyst for various reactions, including the hydrogen evolution reaction [28]. However, EEBP was shown to exhibit too weak of an adsorption of key HER intermediates, which leads to unfavorable pathways of catalytic reactions and greatly complicates its practical application. An effective solution to this problem is the synthesis of heterostructures based on EEBP, among which the greatest attention is paid to the modification of EEBP by functional groups, as well as the decoration of EEBP surface with various catalytically active nanoparticles, for instance, metal atoms and oxides, phosphides, etc. In this respect, EEBP also prevents the agglomeration of catalytically active nanoparticles and leads to an increase in the ECSA of the catalyst, which is undoubtedly important in terms of improving the productivity of the electrolyzer. However, it should be noted that despite all the advantages, EEBP possesses drawbacks, the main one being low electronic conductivity and stability [24,29]. Taking into account all the mentioned peculiarities typical of EEBP and cobalt phosphides, the best way to overcome their shortcomings is to prepare their composites with few-layer graphene structures (FLGSs) because, as is widely known, FLGSs have a large specific surface area and are high electrical and thermal conductivity, strong, often chemical and electrochemical inert and, importantly, low cost. Moreover, the use of nitrogen-doped FLGS (N-FLGS) is considered to be an effective way to redistribute the charge density in the material which leads to the improved adsorption capacity of key HER intermediates. As shown in [30], combining EEBP and N-FLGS is



important to increase activity and stability of the composite catalyst towards HER. Inspired by this idea, we decided to synthesize a nanocomposite containing N-FLGS and EEBP decorated with cobalt phosphide nanoparticles.

In the present study, we propose a simple and effective method of synthesis of a few-layer graphene and phosphorene structure nanocomposite decorated with cobalt phosphide nanoparticles, hereinafter referred to as CoP/EEBP/N-FLGS. It demonstrates high electrocatalytic activity and stability towards HER.

## 2. Materials and Methods

### 2.1. Materials

Sodium nitrate (NaNO$_3$, 99%), melamine (C$_3$H$_6$N$_6$, 99%), graphite rode, red phosphorus (P, 97%), N,N-dimethylformamide (DMF, 99.8%), tetrabutylammonium hexafluorophosphate (Bu$_4$NPF$_6$, 98%), cobalt (II) chloride (CoCl$_2$·6H$_2$O, 98%), and Nafion (5 wt. %) were obtained from Sigma-Aldrich. All reagents were used as received and without further purification.

### 2.2. Preparation of Electrochemically Exfoliated Black Phosphorus (EEBP)

Black phosphorus powder was obtained from red phosphorus using the method described in detail in our previous work [31]. Further, BP electrodes were made by pressing. The electrochemical expansion of BP was carried out in a three-electrode cell with a volume of 10 mL (see Figure S1) containing a solution that was saturated with Ar beforehand and during the experiment. A graphite rod was used as an anode, and Ag/AgCl (sat. KCl) was used as a reference electrode. The BP electrode with a weight of 10 mg was polarized at −7 V for 10 min in a 0.025 M Bu$_4$NPF$_6$ solution in DMF. As a result, a BP electrode was partly destroyed and the rest of the expanded BP was dispersed in the working solution via ultrasonication for 3 min. The EEBP obtained was purified of electrolyte traces by washing with 96% ethanol 4–5 times on a track membrane and vacuum dried at 60 °C for 6 h. Finally, an EEBP powder was obtained.

### 2.3. Preparation of CoP/EEBP

5 mg of EEBP was introduced into a 0.0015 M CoCl$_2$ solution in DMF, then saturated with Ar and subjected to mild ultrasonication for 15 min. The resulting suspension was then transferred to an autoclave purged with Ar. Solvothermal synthesis was carried out at 165 °C for 3 h. At the end, after cooling the autoclave to the room temperature, the suspension was purified of traces of electrolyte by washing with 96% ethanol 4–5 times and vacuum dried at 60 °C for 6 h. Thus, a CoP/EEBP powder was obtained.

### 2.4. Preparation of CoP/EEBP/N-FLGS

Few-layer graphene structures doped with nitrogen atoms were obtained by the plasma-assisted electrochemical exfoliation of graphite carried out in a 1 M NaNO$_3$ + 0.01 M melamine solution as a result of alternate voltage impulses supplied to the graphite electrodes; pulse voltage, duration time and rise time were −180 V (cathodic pulse) and 300 V (anodic pulse), 10 ms and ca. 0.5 μs, respectively [32]. A more detailed description of the set-up and the main physical and chemical processes occurring during the formation of electrolytic plasma are given in [33]. Figure S2 illustrates the nanocomposite synthesis procedure. 5 mg of N-FLGS and 5 mg of EEBP (for more experimental details, see Supplementary Materials) were first unaerated by Ar bubbling 0.0015 M CoCl$_2$ solution in DMF. The mixture was subjected to mild ultrasonication and then held in the autoclave at 165 °C for 3 h. After cooling the autoclave to the room temperature, the suspension was separated from traces of electrolyte, washed with absolute ethanol 4–5 times and vacuum dried at 60 °C for 6 h.

### 2.5. Characterization

Samples for scanning electron microscopy (SEM) and X-ray photoelectron spectroscopy (XPS) were prepared by drop-casting the ultrasonicated suspension of CoP/EEBP/N-FLGS nanocomposite onto the surface of a silicon substrate followed by air-drying at room temperature. SEM images were obtained using a Zeiss SUPRA 25 microscope (Carl Zeiss, Oberkochen, Germany). XPS spectra were registered using a Specs PHOIBOS 150 MCD electron spectrometer (Specs, Berlin, Germany) with a Mg cathode (hν = 1253.6 eV). The vacuum in the spectrometer chamber did not exceed $4 \cdot 10^{-8}$ Pa. A mode of constant transmission energy (40 eV for survey spectra and 10 eV for individual lines) was used to gather the spectra. The survey spectrum was recorded in 1.00 eV increments, while the spectra of individual lines were recorded in 0.03 eV increments. The Shirley method was utilized for background subtraction, and spectra deconvolution was performed using CasaXPS processing software (version 2.3.19). The quantification of atomic content was carried out on the basis of sensitivity factors from the elemental library of CasaXPS. The studied area was 300–700 mm$^2$, and the information depth was 1–2 nm. Transmission electron microscopy (TEM) images were obtained using a JEM-2100 microscope operating at 200 kV. XRD pattern was recorded using an Aeris (Malvern PANalytical B.V., Almelo, The Netherlands) XRD powder diffractometer with a Cu Kα radiation (λ = 1.5406 Å).

### 2.6. Electrochemical Measurements

The linear-sweep voltammetry (LSV) was performed in a three-electrode cell using the setup with a RRDE-3A rotating ring-disk electrode (ALS Co., Ltd., Osaka, Japan) and an Elins P-20X potentiostat (Elins, Chernogolovka, Russia). All measurements were carried out in an Ar-saturated 0.1 M KOH solution at a potential scan rate of v = 10 mV s$^{-1}$. A platinum coil with a 2 cm$^2$ surface area was an auxiliary electrode; Ag/AgCl (saturated KCl) served as a reference electrode. All potential values € were referenced to the Reversible Hydrogen Electrode (RHE) according to the equation: $E_{RHE} = E_{Ag/AgCl} + 0.198 + 0.059 \cdot pH$. The glassy carbon (GC) disc, which was 3 mm in diameter, pressed into a PEEK polymer was used as a working electrode. The surface of the initial GC electrode was polished with a 0.3 μm Al$_2$O$_3$ powder. 2 mg of electrocatalyst was dispersed in a mixture of 0.8 mL of ethanol, 0.1 mL of H$_2$O and 0.1 mL of 1 wt. % Nafion polymer solution by ultrasonication for about 15 min till formation of homogenous ink. Then, 6 μL of the catalyst suspension was drop-cast onto the GC electrode and dried at ambient temperature. As a result, the catalyst loading was ca. 170 μg cm$^{-2}$. The Tafel slopes were calculated according to the equation:

$$\eta = a + b \cdot \log j, \tag{1}$$

where $\eta$ is the overpotential, $a$ is the Tafel constant, $b$ is the Tafel slope, and $j$ is the current density.

The durability of the catalyst was studied via an accelerated durability test (ADT) performed as a potential cycling at a scan rate of 50 mV s$^{-1}$ in the $E$ region from 0.2 to −0.3 V in Ar-saturated 0.1 M KOH. The LSVs for CoP/EEBP/N-FLGS were collected before and after 1000 cycles of potential scanning. All electrochemical measurements were performed at room temperature.

## 3. Results and Discussion

The crystal structure of CoP/EEBP/N-FLGS was characterized via X-ray diffraction. Figure 1 shows the XRD pattern of the CoP/EEBP/N-FLGS nanocomposite as well as the standard cards for CoP (PDF N°01-089-2598), BP (PDF N°01-074-1878) and graphite (PDF N°00-056-0159). The diffraction peak at 26.5° corresponded to the (002) crystal plane of graphite (PDF N°00-056-0159, space group: P63/mmc, cell parameters: $a$ = 2.462 Å, $b$ = 2.462 Å, $c$ = 6.711 Å, α = 90°, β = 90°, γ = 120°), which confirms the presence of few-layer graphene structures [34]. The presence of orthorhombic CoP in the composite is confirmed by clearly defined diffraction peaks located at 31.6°, 46.3°, 48.3°, 52.4°, 56.0°

and 77.1° corresponding to the (011), (112), (211), (103), (020) and (222) crystal planes of CoP (PDF N°01-089-2598, space group: Pnma, cell parameters: $a$ = 5.077 Å, $b$ = 3.281 Å, $c$ = 5.587 Å, $\alpha$ = 90°, $\beta$ = 90°, $\gamma$ = 90°). Diffraction peaks at 16.9°, 35.0°, 56.3° and 66.8° correspond to (002), (111), (016) and (204) crystal planes of orthorhombic BP (PDF №01-074-1878, space group: Bmab, cell parameters: $a$ = 3.32 Å, $b$ = 4.39 Å, $c$ = 10.52 Å, $\alpha$ = 90°, $\beta$ = 90°, $\gamma$ = 90°). It should be emphasized that no peaks typical of cobalt, cobalt oxides or other phases of cobalt phosphide can be found in the diffraction pattern. Thus, the results of XRD analysis indicate the successful formation of graphene–phosphorene structures decorated with CoP nanoparticles. As is known, the presence of CoP favorably affects the activity of composite catalysts [5,35]. According to the literature data, the mechanism of CoP formation on the EEBP surface can be explained as follows: During BP exfoliation, a large number of defects are formed at the edges, resulting in P atoms with lower coordination and a larger number of lone-electrons and, accordingly, a higher reducing ability [36]. As a result, $Co^{2+}$ ions can be reduced to Co atoms at defect positions and then, probably, bond with adjacent P atoms and form phosphides [37].

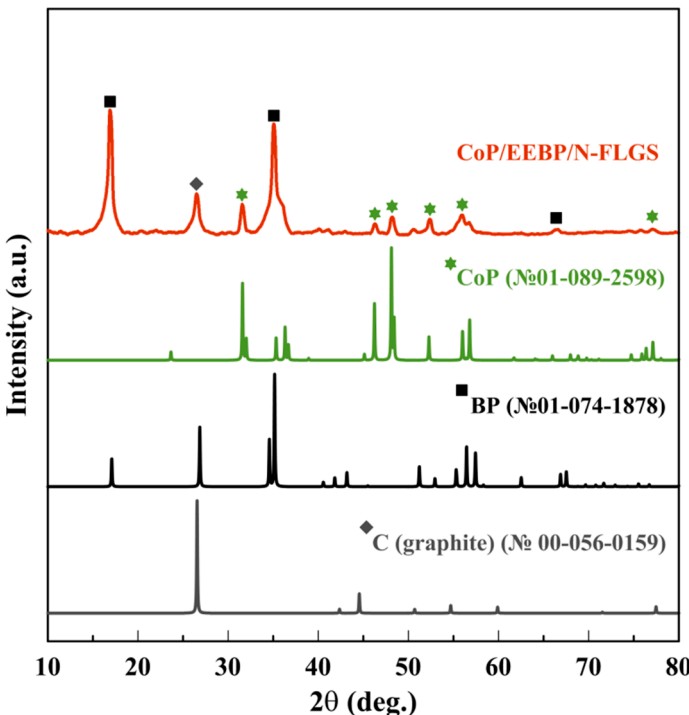

**Figure 1.** XRD pattern of the synthesized CoP/EEBP/N-FLGS nanocomposite and standard reference data for CoP, BP and graphite from PDF cards N°01-089-2598, N°01-074-1878 and N°00-056-0159, respectively.

The surface morphology of the CoP/EEBP/N-FLGS nanocomposite was studied in detail using SEM and TEM techniques as well as related analysis methods. As can be seen from the SEM and TEM images, the sample is an aggregate of typical few-layer graphene and phosphorene structures with a lateral size of several microns (Figure 2a) and a thickness of about 2–5 nm (ca. 6–15 layers) in the case of graphene structures and 2–7 nm (ca. 6–20 layers) in the case of phosphorene structures (Figure 2b), the latter being uniformly covered with nanocrystalline spherical particles that are 2–5 nm in diameter (Figure 2c). The TEM images in Figure 2b,c clearly show crystal lattices with an interplanar spacing of 0.34 nm, 0.52 nm and 0.28 nm, which can be assigned to the (002) crystal plane of graphite, (002) crystal plane of BP and (011) crystal plane of CoP, respectively.

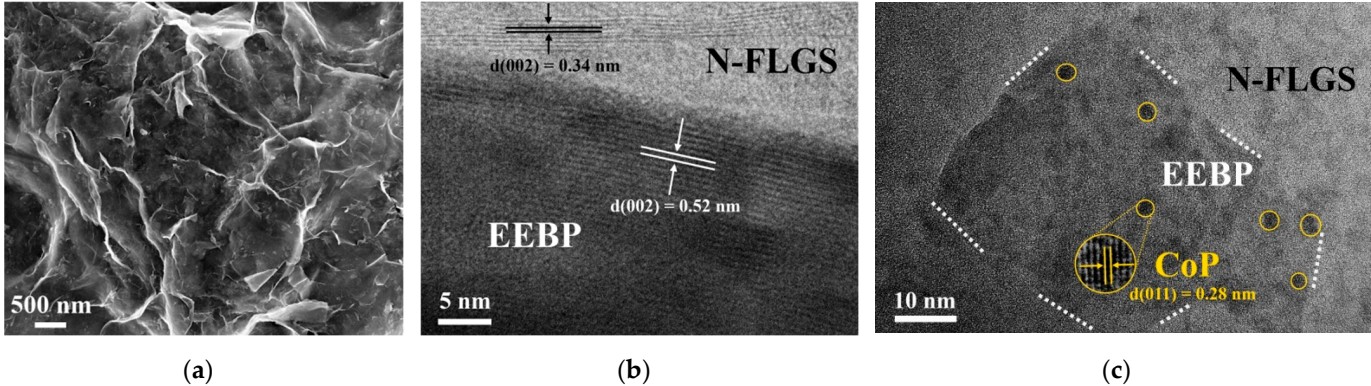

**Figure 2.** (**a**) SEM and (**b**,**c**) TEM images of the CoP/EEBP/N-FLGS nanocomposite.

According to the survey XPS spectrum (Figure S3), CoP/EEBP/N-FLGS nanocomposite contains five elements, namely C, O, N, P and Co, whose surface concentration is given in Table S1. C 1s, Co 2p and P 2p high-resolution spectra are depicted in Figure S4 and Figure 3a,b, respectively. The deconvolution of the C 1s spectrum indicates the presence of various oxygen-containing functional groups on the surface of the graphene substrate, mainly, hydroxyl and epoxy (C–OH/C–O–C) groups [38]. Significant level of FLGS doping with nitrogen atoms during the plasma electrochemical synthesis was demonstrated earlier in [34]. The approximation of the Co 2p spectrum with mixed Gaussian/Lorentzian functions shows the presence of two doublets attributed to Co $2p_{3/2}$ and Co $2p_{1/2}$ as well as satellite peaks at 786.3 and 803.4 eV (Figure 3a). According to the literature data, the peaks at 780.4 eV and 795.7 eV are due to the main photoelectron responses of $Co^{3+}$ ions, while the peaks at 781.8 eV and 797.6 eV are due to $Co^{2+}$ ions [39,40]. Three peaks are observed in the P 2p spectrum at 129.5, 130.4 and 133.1 eV and can be attributed, respectively, to P $2p_{3/2}$, P $2p_{1/2}$ and oxidized phosphorus formed on the nanocomposite surface due to its contact with air [41]. It should be noted that N-FLGS was first to be used to improve the stability of the resulting catalyst. Moreover, according to [30], in graphene–phosphorene structures, EEBP acts as an electron acceptor, while N-FLGS act as an electron donor. In other words, EEBP attracts electrons from N-FLGS, creating a negative charge on the EEBP surface as well as a positive charge on the hole-rich surface of N-FLGS. This also agrees with the earlier conclusion about CoP formation during the interaction of $Co^{2+}$ ions with negatively charged surface of EEBP. Moreover, based on the obtained binding energies of P $2p_{3/2}$ and Co $2p_{3/2}$ peaks, one can also conclude that P has a small negative charge $\delta^-$, while Co has a small positive charge $\delta^+$. This is explained by the fact that the peak in the P $2p_{3/2}$ spectrum at 129.5 eV can be assigned to the P in phosphide [42] since this peak is located at a lower binding energy than that in elemental phosphorus (130.2 eV) [43,44]. At the same time, the Co $2p_{3/2}$ peak at 780.4 eV can be attributed to the binding energy of the cobalt in phosphide since the peak has a positive shift compared to the peak typical of metallic cobalt (778.1 eV) [43,45]. All these facts prove the formation of cobalt phosphide nanoparticles on the surface of EEBP.

The electrocatalytic activity of CoP/EEBP/N-FLGS nanocomposite towards HER was studied in a 0.1 M KOH solution. Figure 4a shows the polarization curves for CoP/EEBP/N-FLGS as well as for EEBP and CoP/EEBP, whose synthesis is described in the Supplementary Materials. As one should expect, at a current density of 10 mA cm$^{-2}$, the CoP/EEBP/N-FLGS nanocomposite demonstrates a substantially lower overpotential of hydrogen evolution reaction, $\eta_{10} \approx 190$ mV, than other samples ($\eta_{10} \approx 865$ mV for EEBP, and $\eta_{10} \approx 315$ mV for CoP/EEBP). The results are summarized in Table 1.

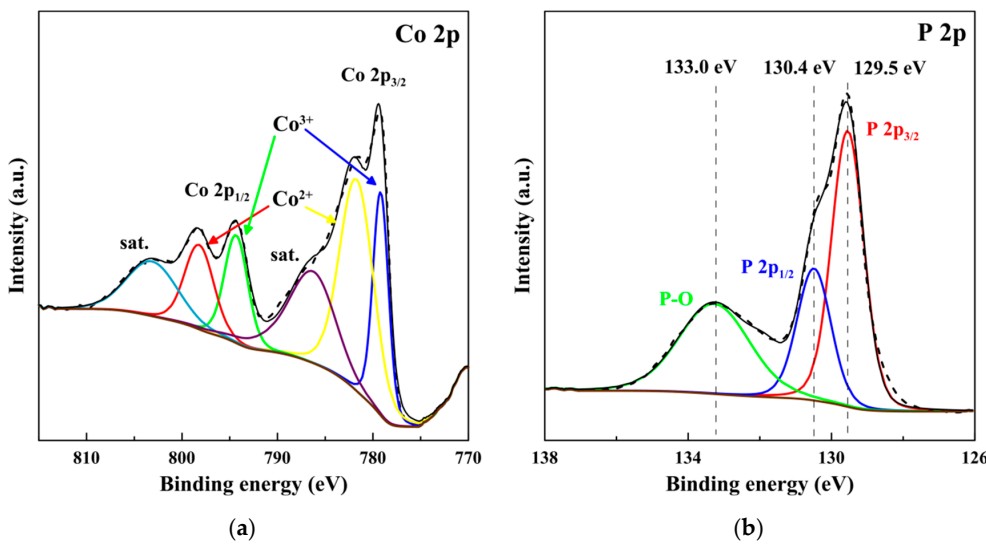

**Figure 3.** High-resolution (**a**) Co 2p and (**b**) P 2p XPS spectra of the CoP/EEBP/N-FLGS nanocomposite.

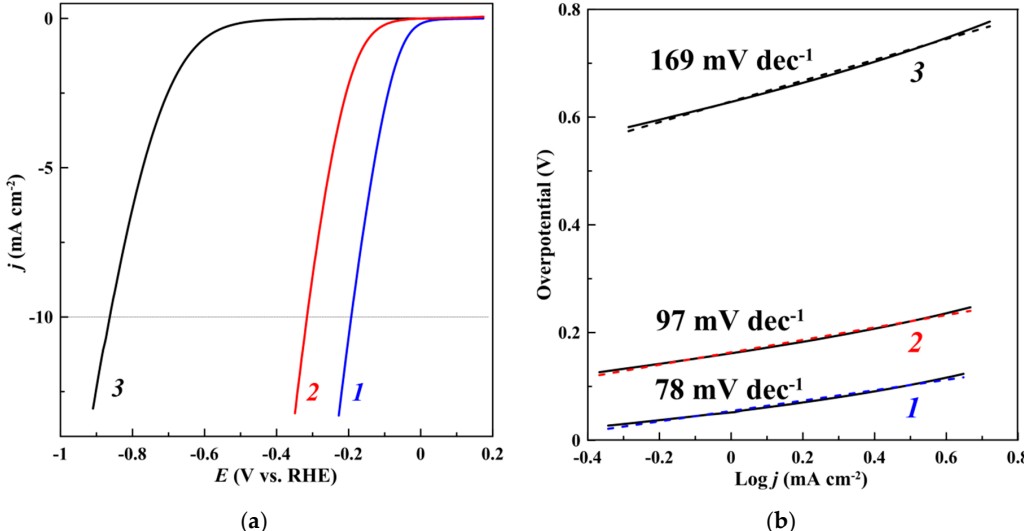

**Figure 4.** (**a**) LSV curves for CoP/EEBP/N-FLGS (1), CoP/EEBP (2) and EEBP (3) measured in a 0.1 M KOH solution ($\nu$ = 10 mV s$^{-1}$, $\omega$ = 2000 rpm; the current density corresponds to the geometric area of the electrode); (**b**) corresponding Tafel plots for CoP/EEBP/N-FLGS (1), CoP/EEBP (2) and EEBP (3).

**Table 1.** Comparison of electrocatalytic activity of synthesized catalysts towards HER.

| Catalyst | $\eta_{10}$ [1], mV | Tafel slope, mV dec$^{-1}$ |
|---|---|---|
| EEBP | 865 | 169 |
| CoP/EEBP | 315 | 97 |
| CoP/EEBP/N-FLGS | 190 | 78 |

[1] $\eta_{10}$ is overpotential at a current density of 10 mA cm$^{-2}$.

Obviously, this fact can be explained by the synergistic effect caused by the combination of N-FLGS and phosphorene structures covered with nanosized particles of cobalt phosphide. Few-layer graphene structures in the composite increase its electrical conductivity and support a more uniform distribution of active centers on the catalyst surface, thus significantly improving the catalytic activity of CoP/EEBP/N-FLGS towards HER [30,46]. In other words, the combination of graphene and phosphorene structures

prevents their agglomeration and leads to an increase in the ECSA of the catalyst. Moreover, CoP nanoparticles located at the edges not only heal defects in EEBP, thus increasing its stability, but also enhance the electrocatalytic performance of composite catalyst. According to Table S1, the overpotential of the hydrogen evolution reaction in alkaline media observed for CoP/EEBP/N-FLGS is noticeably lower compared to the published results for catalysts based on cobalt phosphides [36,47–55]. That is, the electrocatalytic activity of CoP/EEBP/N-FLGS is higher than that of similar catalysts.

The Tafel plots for the studied catalysts are shown in Figure 4b. For CoP/EEBP/N-FLGS, the Tafel slope is 78 mV dec$^{-1}$, while for EEBP and CoP/EEBP, the values are 169 mV dec$^{-1}$ and 97 mV dec$^{-1}$, respectively. The lowest value of the Tafel slope also indicates the best catalytic performance of CoP/EEBP/N-FLGS. The hydrogen evolution reaction is generally accepted to proceed on the catalyst surface either by the Volmer–Tafel mechanism or by the Volmer–Heyrovsky mechanism [56,57]. In alkaline media, the Volmer step is the reduction of a water molecule adsorbed on the catalyst surface alongside an adsorbed hydrogen atom (H$_{ads}$) and a negatively charged hydroxide anion (2). Then, two H$_{ads}$ atoms can either recombine (3) with the formation of a hydrogen molecule leaving the surface (Tafel step), or an H$_{ads}$ atom interacts with the water molecule, producing a hydrogen molecule and OH$^-$ (4, Heyrovsky step).

$$H_2O + e^- \rightarrow H_{ads} + OH^- \tag{2}$$

$$H_{ads} + H_{ads} \rightarrow H_2 \tag{3}$$

$$H_{ads} + H_2O + e^- \rightarrow H_2 + OH^- \tag{4}$$

Based on the Tafel slope value, one can draw a definite conclusion about the rate-determining step of HER. However, this value for CoP/EEBP/N-FLGS (78 mV dec$^{-1}$) is between the values typical of the Volmer (ca. 120 mV dec$^{-1}$) and Heyrovsky (ca. 40 mV dec$^{-1}$) steps [58,59]. For this reason, further studies are required to establish the most realistic HER mechanism for CoP/EEBP/N-FLGS.

An important characteristic of any catalyst is its durability. Figure 5 shows polarization curves for CoP/EEBP/N-FLGS measured before and after 1000 ADT cycles. As can be seen, the overpotential of HER increased only by ca. 10 mV after ADT test, indicating the acceptable long-term stability of the synthesized catalyst in alkaline media.

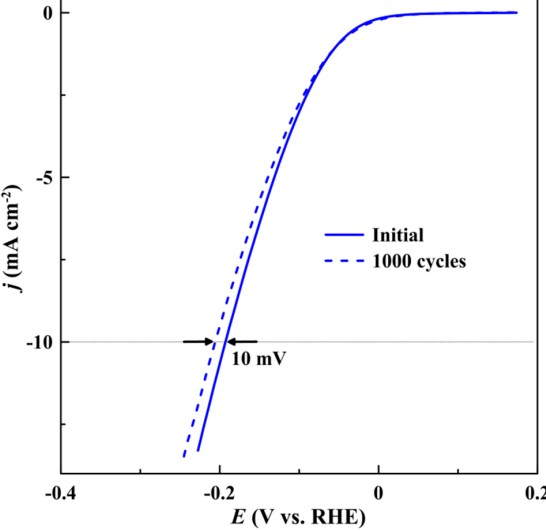

**Figure 5.** LSV curves for CoP/EEBP/N-FLGS before and after 1000 cycles.

## 4. Conclusions

For the first time, we propose a simple and efficient method of synthesis of a CoP/EEBP/N-FLGS nanocomposite, which is an electrochemically exfoliated black phosphorus decorated with CoP nanoparticles with a size of 2–5 nm and stabilized with few-layer graphene structures doped with nitrogen atoms. The CoP/EEBP/N-FLGS nanocomposite demonstrates excellent electrocatalytic performance in HER with low overpotential (ca. 190 mV), low Tafel slope (78 mV $dec^{-1}$) and good stability. The obtained results are due to the synergistic effect caused by the combination of graphene and phosphorene structures, whose surface is decorated with CoP nanosized particles. Thus, the reported CoP/EEBP/N-FLGS nanocomposite is a promising electrocatalyst of the hydrogen evolution reaction, and its simple synthesis opens up broad opportunities for its large-scale production.

**Supplementary Materials:** The following supporting information can be downloaded at: https://www.mdpi.com/article/10.3390/jcs7080328/s1, Figure S1: Schematic diagram of the cell for the electrochemical expansion of black phosphorus; Figure S2: Schematic diagram of the synthesis of CoP/EEBP/N-FLGS nanocomposite; Figure S3: Survey XPS spectrum of CoP/EEBP/N-FLGS nanocomposite; Figure S4: High-resolution C 1s XPS spectrum of CoP/EEBP/N-FLGS nanocomposite; Table S1: Elemental composition of the CoP/EEBP/N-FLGS surface (according to XPS results); Table S2: Comparison of the electrocatalytic activity of catalysts based on cobalt phosphides towards HER.

**Author Contributions:** Conceptualization, V.K.K., R.A.M. and A.G.K.; methodology, V.K.K. and A.S.K.; software, R.A.M.; validation, I.I.K. and A.G.K.; formal analysis, E.N.K. and I.I.K.; investigation, V.K.K.; resources, A.S.K.; data curation, E.N.K. and A.G.K.; writing—original draft preparation, V.K.K. and R.A.M.; writing—review and editing, R.A.M. and A.G.K.; visualization, V.K.K.; supervision, A.G.K.; All authors have read and agreed to the published version of the manuscript.

**Funding:** This research was funded by the Russian Science Foundation, grant number 22-23-00774.

**Data Availability Statement:** The data presented in this study are available on request from the corresponding author.

**Acknowledgments:** This study is supported by the Russian Science Foundation (grant number 22-23-00774) and performed using the equipment of the Multi-User Analytical Center of FRC PCP MC RAS and Multi-User Center of Institute of Solid-State Physics RAS in the frame of state tasks (AAAA-A19-119061890019-5).

**Conflicts of Interest:** The authors declare no conflict of interest.

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
