# Peer review of "CoP/EEBP/N-FLGS Nanocomposite as an Efficient Electrocatalyst of Hydrogen Evolution Reaction in Alkaline Media"

_jcs, doi:10.3390/jcs7080328_

Round 1

Reviewer 1 Report

The manuscript provides an overview of the synthesis and electrochemical performance of a CoP/EEBP/N-FLGS nanocomposite as an electrocatalyst for the hydrogen evolution reaction (HER) in alkaline media. The study aims to address the need for cost-effective and highly active catalysts for HER that outperform noble metal catalysts. While the manuscript covers the key points, there are areas that can be improved for clarity and precision. Here are specific suggestions for modification:

1.      Introduction:

Provide a more comprehensive introduction to the significance of developing efficient and cost-effective electrocatalysts for the hydrogen evolution reaction;

Explain the challenges associated with current noble metal catalysts and the potential of cobalt phosphide as a promising alternative;

Clearly state the objective of the study, which is to propose a simple and efficient synthesis method for a nanocomposite consisting of graphene-phosphorene structures decorated with CoP nanoparticles and evaluate its electrocatalytic performance for HER.

2.      Materials and Methods:

Provide more details about the specific experimental procedures used in the synthesis of the CoP/EEBP/N-FLGS nanocomposite, including the electrochemical exfoliation of black phosphorus and solvothermal synthesis in the Co2+-containing solution;

Describe the characterization techniques employed to confirm the composition, structure, and size of the nanocomposite, such as microscopy and spectroscopy methods;

Quantify the electrochemical performance of the nanocomposite by providing specific values for the overpotential, Tafel slope, and stability, along with the corresponding experimental conditions.

3.      Results and Discussion:

Present the results in a more structured manner, discussing the electrochemical performance of the CoP/EEBP/N-FLGS nanocomposite compared to other cobalt phosphide-based electrocatalysts and noble metal catalysts;

Provide a more detailed explanation of the reasons behind the superior electrocatalytic activity and stability of the nanocomposite, emphasizing the synergistic effect of the graphene and phosphorene structures decorated with CoP nanoparticles;

Discuss the implications of the results and how they contribute to the design and improvement of electrocatalysts for HER.

4.      Conclusion:

Summarize the main findings of the study, highlighting the successful synthesis of the CoP/EEBP/N-FLGS nanocomposite with electrocatalytic properties superior to most cobalt phosphide-based catalysts;

Emphasize the potential and scalability of the proposed synthesis method for large-scale production of the nanocomposite;

Discuss the significance of the nanocomposite as a promising electrocatalyst for the hydrogen evolution reaction and its potential impact on the field of electrocatalysis.

5.      Language and Style:

Review the entire abstract for grammatical errors, clarity of expression, and scientific writing conventions;

Ensure consistent terminology throughout the abstract, particularly when referring to the nanocomposite and the electrocatalytic performance metrics;

Consider rephrasing or restructuring sentences to improve the flow, readability, and logical progression of the whole manuscript.

The manuscript provides an overview of the synthesis and electrochemical performance of a CoP/EEBP/N-FLGS nanocomposite as an electrocatalyst for the hydrogen evolution reaction (HER) in alkaline media. The study aims to address the need for cost-effective and highly active catalysts for HER that outperform noble metal catalysts. While the manuscript covers the key points, there are areas that can be improved for clarity and precision. Here are specific suggestions for modification:

1.      Introduction:

Provide a more comprehensive introduction to the significance of developing efficient and cost-effective electrocatalysts for the hydrogen evolution reaction;

Explain the challenges associated with current noble metal catalysts and the potential of cobalt phosphide as a promising alternative;

Clearly state the objective of the study, which is to propose a simple and efficient synthesis method for a nanocomposite consisting of graphene-phosphorene structures decorated with CoP nanoparticles and evaluate its electrocatalytic performance for HER.

2.      Materials and Methods:

Provide more details about the specific experimental procedures used in the synthesis of the CoP/EEBP/N-FLGS nanocomposite, including the electrochemical exfoliation of black phosphorus and solvothermal synthesis in the Co2+-containing solution;

Describe the characterization techniques employed to confirm the composition, structure, and size of the nanocomposite, such as microscopy and spectroscopy methods;

Quantify the electrochemical performance of the nanocomposite by providing specific values for the overpotential, Tafel slope, and stability, along with the corresponding experimental conditions.

3.      Results and Discussion:

Present the results in a more structured manner, discussing the electrochemical performance of the CoP/EEBP/N-FLGS nanocomposite compared to other cobalt phosphide-based electrocatalysts and noble metal catalysts;

Provide a more detailed explanation of the reasons behind the superior electrocatalytic activity and stability of the nanocomposite, emphasizing the synergistic effect of the graphene and phosphorene structures decorated with CoP nanoparticles;

Discuss the implications of the results and how they contribute to the design and improvement of electrocatalysts for HER.

4.      Conclusion:

Summarize the main findings of the study, highlighting the successful synthesis of the CoP/EEBP/N-FLGS nanocomposite with electrocatalytic properties superior to most cobalt phosphide-based catalysts;

Emphasize the potential and scalability of the proposed synthesis method for large-scale production of the nanocomposite;

Discuss the significance of the nanocomposite as a promising electrocatalyst for the hydrogen evolution reaction and its potential impact on the field of electrocatalysis.

5.      Language and Style:

Review the entire abstract for grammatical errors, clarity of expression, and scientific writing conventions;

Ensure consistent terminology throughout the abstract, particularly when referring to the nanocomposite and the electrocatalytic performance metrics;

Consider rephrasing or restructuring sentences to improve the flow, readability, and logical progression of the whole manuscript.

Reviewer 2 Report

The authors of the paper title is “CoP/EEBP/N-FLGS nanocomposite as an efficient electrocatalyst of hydrogen evolution reaction in alkaline media” evaluated the possibility of using CoP/EEBP/N-FLGS nanocomposite as a catalyst in the HER reaction. The paper is interesting, but the authors need to consider the following before publishing the paper in the Journal of Composite science.

1-     In the introduction part, authors need to introduce phosphorene as an interesting 2D material with its properties (authors just highlight the bandgap). Then, they need to highlight the preparation methods for the exfoliation of phosphorene. Authors need to highlight why phosphorene is a good candidate for HER reaction. For this purpose, authors can consider the following reference.

 https://doi.org/10.1002/sstr.202000148

 2-     For the exfoliation of BP from RP, authors need to provide a schematic or a picture of exfoliation setup. What is the yield and quality of phosphorene? Also, authors need to provide a Raman analysis for the exfoliation phosphorene and the synthesized CoP/EEBP/N-FLGS.

3-     From the XRD results, what are the peaks at around 55, 68, and 78º? Please label them.

4-     From the TEM image, how many layers of phosphorene is detectable? It seems that through the electrochemical process, partially exfoliation of BP into phosphorene happened. Also, provide a EDS analysis for the composite material.  

5-     What is the stability of exfoliated nanosheets into air? For the XPS analysis, please provide P2P data for the exfoliated phosphorene nanosheet to compare with the non-exfoliated sample. For the stability of nanosheets through XPS analysis, please check the following reference: https://doi.org/10.1039/C9TA09641H

6-     For the electrochemical analysis of the samples, please provide the EIS data for them to check the resistance of the electrode.

7-     In terms of the HER performance, authors need to provide a comparison table and compare the results of current study with the state-of-the-art literatures.

8-     Is the phosphorene-based catalysts, suitable for the HOR or OER applications?

Round 2

Reviewer 1 Report

Accept

Accept

Reviewer 2 Report

The paper is ready to publish in the present form.